# Heterologous Expression of an Insecticidal Peptide Obtained from the Transcriptome of the Colombian Spider *Phoneutria depilate*

**DOI:** 10.3390/toxins15070436

**Published:** 2023-07-02

**Authors:** Julieta Vásquez-Escobar, Dora María Benjumea-Gutiérrez, Carolina Lopera, Herlinda C. Clement, Damaris I. Bolaños, Jorge Luis Higuita-Castro, Gerardo A. Corzo, Ligia Luz Corrales-Garcia

**Affiliations:** 1Grupo de Toxinología y Alternativas Farmacéuticas y Alimentarias, Facultad de Ciencias Farmacéuticas y Alimentarias, Universidad de Antioquia, Medellin 1226, Colombia; dora.benjumea@udea.edu.co (D.M.B.-G.); carolina.loperal@udea.edu.co (C.L.); 2Departamento de Medicina Molecular y Bioprocesos, Instituto de Biotecnología, Universidad Nacional Autónoma de México, A.P. 510-3, Cuernavaca 62250, Mexico; herlinda.clement@ibt.unam.mx (H.C.C.); dama104ris@gmail.com (D.I.B.); corzo@ibt.unam.mx (G.A.C.); 3PECET—Programa para el Estudio y Control de Enfermedades Tropicales, Facultad de Medicina, Universidad de Antioquia, Medellín 050010, Colombia; jluis.higuita@udea.edu.co; 4Departamento de Alimentos, Facultad de Ciencias Farmacéuticas y Alimentarias, Universidad de Antioquia, Medellín 1226, Colombia

**Keywords:** insecticidal, heterologous, *Phoneutria*, spider, venom

## Abstract

Spider venoms are composed, among other substances, of peptide toxins whose selectivity for certain physiological targets has made them powerful tools for applications such as bioinsecticides, analgesics, antiarrhythmics, antibacterials, antifungals and antimalarials, among others. Bioinsecticides are an environmentally friendly alternative to conventional agrochemicals. In this paper, the primary structure of an insecticidal peptide was obtained from the venom gland transcriptome of the ctenid spider *Phoneutria depilata* (Transcript ID PhdNtxNav24). The peptide contains 53 amino acids, including 10 Cys residues that form 5 disulfide bonds. Using the amino acid sequence of such peptide, a synthetic gene was constructed de novo by overlapping PCRs and cloned into an expression vector. A recombinant peptide, named delta-ctenitoxin (rCtx-4), was obtained. It was expressed, folded, purified and validated using mass spectrometry (7994.61 Da). The insecticidal activity of rCtx-4 was demonstrated through intrathoracic injection in crickets (LD_50_ 1.2 μg/g insect) and it was not toxic to mice. rCtx-4 is a potential bioinsecticide that could have a broad spectrum of applications in agriculture.

## 1. Introduction

Some arthropods, such as centipedes, scorpions, spiders and wasps, use venom to incapacitate their prey. Except for predatory beetles, spiders are the most successful insecticidal animals. They are the most abundant and widespread arachnids, with nearly 50,000 existing species described to date [1,2]. The success of the spiders relies, in part, on the effectiveness of their venom, which is designed to paralyze and kill prey or predators as quickly as possible with a complex of enzymes, neurotoxins and cytolytic compounds. Most spider venoms consist of small disulfide-rich peptide neurotoxins, the largest and most extensively studied group of spider toxins [1]. These neurotoxins rapidly alter ion conductance (ion channel toxins) and, to a lesser extent, affect neurotransmitter exocytosis (presynaptic toxins). Many of these spider peptide toxins are insecticidal. Yet, because of molecular serendipity, some insecticidal peptides are also toxic to mammals. In particular, insect-selective toxins have been patented for their potential use as bioinsecticidal agents to control phytophagous pests or insect vectors [3].

Approximately 648 peptide toxins from 84 spider species have been described and reviewed in UniProtKB (www.uniprot.org, accessed on 21 April 2023), a curated database with available information on peptides and proteins from various animals, including 150 peptides that have been associated with insecticidal activity [4]. In recent years, it has become clear that spider venoms are much more complex than previously thought, with some venoms containing more than 1000 unique, small, disulfide-rich peptides [5]. Using a conservative estimate of 50,000 existing species and more than 200 peptides per venom, spider venoms may contain over 10 million bioactive peptides [6]. Still, less than 0.01% of this proteomic diversity has been explored to date [7,8].

Spiders belonging to the genus *Phoneutria* possess potent neurotoxic venoms, making them some of the world’s most biotechnologically important spider venoms. Roughly 400 peptides and proteins, ranging from 1.2 to 27 kDa, have been isolated from the venom of *Phoneutria* species, but the complete or partial amino acid sequence has been determined for only circa 100 peptides. Also, the epidemiology of bites from these spiders has been registered [9,10]. The most studied venom peptides have been isolated from the species *P. nigriventer*, *P. reidyi*, *P. keyserlingi* and *P. salei*. The spider *P. depilata* (formerly *P. boliviensis*) is a poorly studied species (Figure 1) [10,11]. The venom peptides of this species also represent a rich source of valuable neurotoxins that could have potential pharmacological and insecticidal leads active toward insects and mammals [12].

In a previous work, the transcriptome of the *P. depilata* was annotated (ENA project PRJEB33730) [13]. Considering the insecticidal potential of the venom of this spider, a transcript search coding for putative primary structures with insecticidal activities was performed using the tblastn tool (https://blast.ncbi.nlm.nih.gov/Blast.cgi (accessed on 22 July 2020)). A transcript, ID PhdNtxNav24, that codes for an insecticidal protein with 60.9% identity to an annotated sodium channel neurotoxin from *Phoneutria* was found. The gene of such insecticidal neurotoxin was synthetically constructed, cloned and recombinantly expressed. Here, the primary structure of a spider peptide, named Ctx-4, is uncovered. rCtx-4 caused paralysis and death in crickets, but it had no effects on mice injected intracranially (ic). We also found the CTX-4 sequence in cDNA obtained from the glands of the spider *P. depilata*, validating the transcriptome.

## 2. Results

### 2.1. Design and Gene Amplification

The *P. depilata* transcriptome yielded the transcript PhdNtxNav24 [13] (Table 1), a transcript that codes for the putative peptide with 60.9% identity to δ-CNTX-Pn1a, a neurotoxin active on sodium channels from *P. nigriventer*. The transcript named Ctx-4 was also selected for recombinant expression because of its e-value (9 × 10^−21^) and its content of cysteines (10), which can form five disulfide bonds typical of insecticidal peptides from the venom of ctenid spiders [14]. Based on the reverse-translated sequence, four oligonucleotides were designed to assemble the Ctx-4 gene through overlapping PCR (Table 2).

### 2.2. Cloning in pQE-30 Expression Vector

The obtained Ctx-4 insert (203 bp) was digested with *Bam*HI and *Pst*I and cloned into the pQE30 vector. The resulting recombinant plasmid was used to transform *Escherichia coli* XL1Blue cells. Colony PCR was used to evaluate seven colonies that showed an amplification band of the expected size (~400 bp). The colonies expressed a protein of the predicted size (~8 kDa) as observed by SDS-PAGE. Colonies 4 and 5 showed the most significant protein expression and they were subjected to plasmid purification and sequencing. The plasmid with the correct sequence was selected for further protein expression pQE30-Ctx-4 (Figure 2).

### 2.3. Expression of Recombinant Ctx-4

*E. coli* Origami and Shuffle cells were transformed with pQE30-Ctx-4 plasmids, respectively. The expression was induced with 0.1 mM IPTG at 16 °C for 12 h and 200 rpm. After induction, the recovered cells were sonicated, then centrifuged to obtain soluble and insoluble fractions and all samples were analyzed using SDS-PAGE (Figure 3). The expressed protein in the insoluble fraction (Figure 3, lane 4) was solubilized with 6M GnCl and purified through nickel affinity chromatography (Figure 3). A sample from this purification was repurified using RP-HPLC (Figure 4A), yielding a broad fraction with several peptide isoforms. One of these isoforms (37.4 min, corresponding to the elution at 32.4% ACN) showed an experimental molecular mass of 7995.7 Da (oxidized) (Figure 4B).

The samples eluted from the affinity chromatography step were prepared for a folding process and purified using RP-HPLC. Three fractions of the recombinant rCtx-4 were obtained with experimental molecular masses of 7998.35, 7994.61 and 7994.20 Da at 36.5, 38.7 and 39.5% ACN, respectively (Figure 5).

### 2.4. Transcriptome Validation: Searching the rCtx-4 Sequence in a Spider Venom Gland

An approach to validate the transcriptome of the spider *P. depilata* was to demonstrate the existence of the coding sequence of rCtx-4 as a transcript from the genetic material of the spider’s venom gland. Therefore, the RNA of just one pair of venom glands was extracted, obtaining 11.4 μg of the total RNA and 4 µg was used to synthesize the corresponding cDNA. Using the designed oligonucleotides (Table 2), a cDNA amplification of the expected size was obtained and it was cloned into the pCR 2.1-TOPO^®^ cloning vector. The plasmid was used to transform *E. coli* XL1Blue quimiocompetent cells. Some cells resulting from the transformation were evaluated using colony PCR. Subsequently, plasmids from four positive colonies were purified and DNA sequenced. All four DNA sequences have an identity >92.4% with the recombinant peptide Ctx-4 (Figure 6A).

Additionally, using the blastn tool, we found a similarity between the sequences of pCR 2.1-TOPO-Ctx plasmids and the PhdNtxNav23 transcript (94.3% identity and 8.2e-76 e-value). By searching for identity with the blastx tool between plasmids’ sequences and the transcriptome, we found the same transcript used for construction of the gene encoding Ctx-4, PhbNtxNav24 transcript showed an even higher similarity (98.1% identity and 2.9e-34 e-value) (Figure 6B).

rCtx-4 peptide is 92.5% identical to the PhdNtxNav23 transcript due to a difference in four amino acids (A1G, R2K, P39A and V44L). In comparison, the sequence of the TOPO/Ctx-4_1 plasmid is 94.3% identical to the transcript due to the difference in three amino acids (A1G, R2K and P39A) (Figure 6A); however, TOPO/Ctx-4 plasmid is 98.1% identical to rCtx-4 and PhbNtxNav24, since they only differ in one amino acid (V44L) (Figure 6B).

### 2.5. Acute Toxicity of Recombinant Peptide Ctx-4 in Mice

None of the mice treated (via ic) at doses of 0.9 μg/g of the folded rCtx-4 peptide showed signs of toxicity. There was no evidence of significant weight variation during the 14 days of the experiment, with an increase in weight at the end of the observation window. These same results were presented with the administration of 1.0 and 1.5 μg/g, showing that the recombinant peptide is not toxic to mice by this route at the used doses. The macroscopic evaluation of the organs revealed no size, shape, color or friability alteration.

### 2.6. Toxicity of the rCtx-4 Peptide in Crickets

An amount of 5 µg of each fraction of the folded rCtx-4 peptide were injected into crickets, producing severe toxicity characterized by paralysis, inability to reverse and tremors in the animal for more than 30 min. Crickets injected with the 36.5% can fraction (Figure 5) showed signs of paralysis 1 minute after administration, whereas injection of the 38.7 and 39.5% ACN fractions resulted in immediate paralysis of the crickets.

Crickets injected with the 36.5% ACN fraction recovered some mobility 2 h after administration and they were fully recovered after 24 h. Additionally, crickets that received 1 and 2.5 μg of the 38.7% ACN fraction exhibited the same severe toxicity as crickets receiving 5 μg of this fraction. Crickets injected with the 38.7% and 39.5% ACN fractions died 29 and 32 h after administration, respectively.

### 2.7. Mean Paralyzing Dose (PD_50_) of the Recombinant Peptide Ctx-4 in Crickets

The determination of the PD50 was evaluated using the 38.7% ACN fraction obtained from the RP-HPLC purification of folded rCtx-4 (Figure 5), with an initial dose of 1.3 μg/g. The result obtained from the PD50 of rCtx-4 was 1.2 μg/g, with a standard error of 0.3 μg/g.

## 3. Discussion

This is the first report of a recombinant insecticidal peptide associated with the venom of *P. depilata*. Our research started with the selection of a promising insecticidal peptide coding from an assembled transcriptome of the venom gland of the ctenid spider *P. depilata* [13]. The selected peptide, Ctx-4, was analyzed and designed for heterologous expression. As expected, the rCtx-4 peptide was expressed in the insoluble fraction, forming inclusion bodies, which are protein aggregates that form because the expressed proteins are toxic to the host, are misfolded, undergo proteolytic degradation or associate with each other [15]. The high cysteine content of rCtx-4 (10 cysteines) probably promotes aggregation by forming non-native disulfide bridges or by clustering hydrophobic regions. However, the formation of inclusion bodies allows the continuous expression of recombinant proteins without compromising the viability of *E. coli* [16]. Other studies have demonstrated the inclusion body formation in the recombinant expression of spider peptides with the same number of cysteines (10 cysteines, 5 disulfide bridges), such as peptides from the venom of the spider *Oxyopes lineatus* [14].

The expression of spider venom proteins in the soluble fraction was achieved for dermonecrotic proteins from *Loxosceles intermedia* venom, which have a high molecular weight and few cysteines in their sequences [17,18], and for the Phα1β peptide (ω-ctenitoxin-Pn4a), of 10,894 Da and 12 cysteines, by adding fusion proteins to the coding sequences [19]. For the heterologous expression of Hainantoxin-IV, a 35-amino acid, 6-cysteine peptide, from the spider *Selenocosmia hainana* venom, the glutathione-S-transferase (GST) tag and a small ubiquitin-related modifier (SUMO) were combined to promote the solubility and folding of the recombinant protein, without the need for subsequent folding processes; however, the recombinant peptide was obtained in both the soluble and insoluble fractions [20]. Although this strategy could be promising for the expression of the recombinant peptide Ctx-4 in the soluble fraction, the yield decreased due to the separation of the fusion protein and the extra purification process [21]. Yields of less than 10% have been reported for other peptides using these strategies [20].

The chromatogram obtained from the RP-HPLC purification of the folded recombinant peptide (Figure 5), in which the three fractions obtained have similar molecular masses, suggests the presence of Cys-Cys isoforms generated during folding by the oxidation of the thiol groups in the formation of the disulfide bonds, as has happened for other cysteine-rich peptides from recombinantly expressed spider venoms [14]. Given the richness of cysteine residues in this recombinant peptide, different isoforms would be feasible during folding [10]; five disulfide bonds can be formed with multiple binding combinations and, consequently, a large variety of isoforms. However, it has been observed that, even after the folding of cysteine-rich spider peptides, not all isoforms obtained through RP-HPLC are biologically active; this is the case of the Ba1 insecticidal peptide from *Brachypelma albiceps*, for which only one of the purified isoforms showed insecticidal activity [22]. The three isoforms obtained after the folding of the rCtx-4 peptide were toxic to crickets, suggesting an effective folding strategy. However, other approaches could be evaluated for a single isoform with biological activity.

The chosen expression model in *E. coli* has been widely used for heterologous expression of spider venom toxins and can be profitable by obtaining large amounts of recombinant peptides (10–20 mg/L), although this yield decreases after folding and purification of the peptide [23]. The amount of rCtx-4 obtained after folding and RP-HPLC purification of the fraction with activity was 1.5 mg/L; although this is not a large amount of recombinant peptide, it is acceptable in this work considering that yields can be reduced by more than 10% of the original product [20]. However, other processes or systems that can improve this performance could be tested.

The alignments of the obtained transcripts, blastn (94.34%) and blastx (98.11%), showed few differences between the sequences, representing isoforms of the gene [24]. In the blastn alignment, we found an identity between the PhbNtxNav23 transcript and the TOPO/Ctx-4 plasmid, but some amino acids did not match (A1G, R2K, P39A). However, for the PhbNtxNav24 transcript, which was used to obtain the initial sequence to construct the rCtx-4 peptide, the difference lies in a single amino acid (leucine (L44) for the transcript and valine (V44) for the TOPO/Ctx-4 plasmid). This amino acid difference is due to a change in two nucleotides in the codon, which can be a gene mutation or a natural isoform. An amino acid mismatch between two sequences may or may not imply a change in peptide function [25]. However, it must be considered that both the venom and the mRNA are dynamic and can change under different circumstances in response to external and internal stimuli; proteins can be post-translationally modified, translocated, synthesized or degraded [26]. However, the finding of a cDNA sequence from one venom gland that is 98.1% identical to a sequence in the transcriptome of another venom gland of *P. depilata* spider leads us to propose that this peptide is present in the venom of this spider and plays an essential role in the capture and envenomation of insects as food.

In experimental animals, the rCtx-4 peptide showed no toxicity to mice (1.5 μg/g, via ic). Although the rCtx-4 sequence is annotated in the transcriptome as a sodium neurotoxin, it may not affect sodium channels in mammals but in invertebrates, which explains its toxicity to crickets but not to mice. However, its effect in mammals could be on a different target, perhaps the central nervous system and/or an effect unrelated to toxicity. Additionally, the peptide δ-ctenitoxin-Pn1a from *P. nigriventer*, which was used to search for putative peptide sequences with an insecticidal activity and from which we obtained the transcript to construct the Ctx-4 gene, did not produce any apparent symptoms of intoxication in mice after intracerebroventricular (icv) administration (dose ~1.5 μg/g) [27].

Given the highly toxic effects of the three-folded isoforms of the rCtx-4 in crickets, we assumed there would be a high similarity; however, the differences may be due to the binding pattern of disulfide bonds formed during folding, which generated different isoforms [14]. The toxicity results of the folded rCtx-4 peptide are comparable to its reference peptide, δ-ctenitoxin-Pn1a from *P. nigriventer*, which is highly toxic to flies (*Musca domestica*) under native conditions (LD_50_ 3.8 ± 2 ng/fly), producing neurotoxic effects, including the inability to reverse, tremors and uncoordinated movements, and to cockroaches (*Periplaneta americana*), knocking them out immediately with complete loss of coordination and abdominal contraction (0.5–2.5 μg/g) [27].

Similar studies have been performed on other spider venom peptides with insecticidal activity. The insecticidal neurotoxin Ba1 from *Brachypelma albiceps* has been heterologously expressed. Its insecticidal activity has been tested, successfully reporting such activity in one of the isoforms obtained after folding. The rapid onset and intense insecticidal activity of this neurotoxin, like the sodium toxins of spider venoms, suggest that its molecular target may be the voltage-gated sodium channels of insects [25]. The peptide δ-ctenitoxin-Pn1a exerts its insecticidal effect by slowing down the inactivation of sodium currents in the central nervous system of insects by binding to voltage-gated sodium channels [28]. Given that the rCtx-4 peptide is associated with a toxin sequence in the *P. depilata* transcriptome that alters sodium channels, we suggest that the rCtx-4 peptide may have the same effect on insect sodium channels since the biological activities are similar in both invertebrates and in mammals [28]. Moreover, some other spider toxins that are lethal to insects and vertebrates, such as δ-atracotoxins, have been shown to act by binding to the S3/S4 loop of the domain IV of the voltage-gated sodium (Na_V_) channels [29].

The PD_50_ determined for the 38.7% ACN fraction of the rCtx-4 peptide in crickets cannot be compared in-depth with the toxic doses found for the reference peptide δ-ctenitoxin-Pn1a because the animal models used were different, presenting a diverse abundance of sodium channels, leading to different sensitivity to peptides. However, we believe that the toxic effect of the rCtx-4 peptide is comparable to that of the peptide from *P. nigriventer*, since the rCtx-4 PD_50_ (1.2 ± 0.2 μg/g) was in the range of the toxic effects observed in cockroaches (0.5–2.5 μg/g) but was higher than that reported for toxicity in flies (3.8 ± 2 ng/fly) [27]. Compared to other toxins affecting insects, we found that rCtx-4 had a lower PD_50_ (1.2 ± 0.2 μg/g) than the rMagi4 peptide (PD_50_ 4.8 ± 1 μg/g), a putative toxin of δ-atracotoxin-Mg1a [29], but higher than the rOsu1 peptide, obtained from the venom of *Oculicosa supermirabilis*, with a PD_50_ of 0.27 μg/g [30]. Thus, the rCtx-4 peptide could be a candidate for further studies of insecticidal activity and a potential bioinsecticide as an environmentally friendly alternative to conventional agrochemicals.

## 4. Conclusions

The recombinant Ctx-4 was shown to have toxic activity in crickets, while there was no toxicity in mice. This result represents a potential agro-industrial use to control insect pests, given that the rCtx-4 peptide can be obtained by recombinant means and is correctly folded. Although the PD_50_ for insects is not the lowest found for recombinant spider toxins, it is appropriate to consider the rCtx-4 peptide as a potential bioinsecticide because the widespread use of chemical insecticides has resulted in genetic selection pressure that has led to the development of insecticide-resistant arthropods, as well as concerns for human health and the environment.

## 5. Materials and Methods

### 5.1. Gene Design and Construction

The transcript PhdNtxNav24 (Table 1) was translated to protein (Figure 7A). Then, such primary structure was reversed-translated to DNA (Figure 7B) and then adjusted and equilibrated considering the preferential codon usage for *E. coli* K12 (Figure 7C) (http://www.kazusa.or.jp/codon (accessed on 15 June 2020)). Afterward, four oligonucleotides were designed (Table 2) containing overlapping ends to construct the Ctx-4 gene de novo. The synthesized oligonucleotides were purified by acrylamide (15%) and urea (7M) gels and the concentration was set at 10 µM.

The Ctx-4 gene was constructed by overlapping the designed oligonucleotides. Briefly, 0.1 µM of Ctx4-Lw2 and Ctx4-Up3 and 0.4 µM of Ctx4-Up1 and Ctx4-Lw4 were mixed with dNTPs (200 µM), polymerase buffer (1X) and 1 U of *Vent* polymerase (New England BioLabs Inc., Ipswich, MA, USA). The reaction solution was amplified for 30 cycles under the following conditions: 94 °C/30 s, 57 °C/30 s and 72 °C/30 s. The final extension step was 72 °C/10 min. The PCR product was run on 1.2% agarose gels containing GelRed^®^ (Biotium, Fremont, CA, USA) and visualized under ultraviolet (uv) light. The amplification product was then excised from the agarose gel, purified with phenol/chloroform and precipitated with a butanol–ethanol solution.

### 5.2. Cloning in pQE30 Expression Vector

The purified gene and the pQE30 expression plasmid were digested with the enzymes *Bam*HI and *Pst*I (New England BioLabs Inc., Ipswich, MA, USA). After digestion, the Ctx-4 gene and the vector were run and purified from an agarose gel. The digested DNAs were mixed in a ratio of 1:5 and then T4 ligase (1 Weiss U) (Thermo-Fisher Co., San José, CA, USA) and T4 DNA ligase buffer (1x) were added. The reaction was incubated overnight at 16 °C. The ligation product (pQE30::Ctx-4) was used to transform chemically competent *E. coli* XL1-Blue cells (Agilent, Santa Clara, CA, USA). Transformed cells were grown on 2xYT agar (17 g/L tryptone, 10 g/L yeast extract, 5 g/L NaCl, 1.5% agar) supplemented with 200 µg/mL of ampicillin (Sigma, St. Louis, MO, USA). After incubation (37 °C, 14 h), some colonies were picked up to assess the presence of recombinant plasmid presence using colony PCR with the oligonucleotides pQE-Fwd and pQE-Rev (Table 2). Positive colonies were seeded in 3 mL of 2xYT, incubated overnight at 37 °C and 180 rpm and then subjected to plasmid purification (High Pure Plasmid Isolation Kit, Roche^®^, Basel, Switzerland). The plasmids were sequenced at the Institute of Biotechnology, UNAM, Mexico City, Mexico.

### 5.3. Expression of the Recombinant Toxin Ctx-4

The nucleotide sequence of the recombinant plasmid pQE30::Ctx-4 was confirmed and it was used to transform *E. coli* Origami and Shuffle cells through heat shock. One colony from each transformation was used to inoculate 3 mL of 2xYT plus ampicillin (200 µg/mL) and cultured overnight at 37 °C, 250 rpm. After 24 h, 1 mL of such culture was used to inoculate 50 mL of 2xYT plus ampicillin and incubated for 4 h at 37 °C, 250 rpm. Finally, the culture was added to 200 mL of 2xTY plus ampicillin and allowed to grow at 37 °C, 250 rpm to an optical density (OD 600nm) of ~1.0 U. At this point, 0.1 mM IPTG (IsoPropyl-ß-1-D Thiogalactopyranoside—Sigma, St. Louis, MO, USA) was added as an inductor and the bacterial culture was further incubated overnight at 16 °C, 250 rpm.

After incubation, cells were recovered using centrifugation at 6000 *g* for 20 min at room temperature (RT). Then, the cells were resuspended in 10 mL of 50 mM Tris-HCl buffer, pH 8. Cells were sonicated for 3 min (5 s intervals of sonication and 5 s rest, at 30% power; Ultrasonicator BioBase, Shandong, China). The lysate was centrifuged (12,300 *g* for 30 min at RT). The insoluble fraction was washed with 20 mL of 50 mM Tris-HCl buffer, pH 8 and then again centrifuged. The pellet containing inclusion bodies was solubilized with guanidine hydrochloride (GdHCl) 6 M, Tris-HCl 50 mM buffer, pH 8. The recombinant Ctx-4 (rCtx-4) was purified from the solubilized inclusion bodies using Ni-NTA agarose (Qiagen, Hilden, Germany) and eluted with imidazole 400 mM. The eluate (6 M GdHCl and 400 mM imidazole) was diluted with 50 mM Tris-HCl, pH 8, to 2 M GdHCl and then a refolding buffer (0.1 M Tris, pH 8, 2 M GdHCl, 1 mM GSSG and 10 mM GSH) (GSSG: Glutathione disulfide; GSH: Glutathione; Sigma Aldrich, St. Louis, MO, USA) was added. The refolding reaction was incubated at 4 °C for 96 h. The recombinant peptide was then purified through analytical RP-HPLC using a Chromaster (Hitachi, China) equipped with a DAD detector and a Zorbax 300SB column (C_18_, 4.6 × 250 mm, Agilent Technologies, Inc., Santa Clara, CA, USA) using an HPLC linear gradient from 0 to 60% acetonitrile (ACN) in 0.1% TFA from minute 5 to minute 65 at 1 mL/min and monitored at 230 nm.

### 5.4. Transcriptome Validation

The venom glands of an adult female spider of *P. depilata* were extracted and then immersed and homogenized in TRIzol^®^ (Thermo-Fisher Co., San José, CA, USA). The RNA was extracted according to the guanidine/phenol isothiocyanate method described by Chomczynski and Sacchi (1987) [31]. The pellet was washed with 75% ethanol in diethylpyrocarbonate (DEPC) water and allowed to dry. The product was suspended in DEPC water for spectrophotometric quantification at λ 260 nm.

The cDNA from 4 μg of extracted total RNA was obtained using the 3’RACE kit (Invitrogen, Carlsbad, CA, USA) according to the manufacturer’s protocol. The gene coding for the recombinant peptide was amplified from the cDNA pool using specific oligonucleotides, which were previously designed from the transcriptome information (cDNA-Ctx-Fwd and cDNA-Ctx-Rev, Table 2). The PCR reaction consisted of 0.4 µM of each oligonucleotide, 200 µM dNTPs, 1X polymerase buffer and 1 U of *Taq* polymerase. The PCR conditions were 94 °C/30 s, 57 °C/30 s and 72 °C/30 s. The final extension step was 72 °C/10 min). The amplified DNA fragment (~150 bp) was purified from the agarose gel using the High Pure PCR Product Purification Kit (Roche^®^, Basel, Switzerland).

According to the manufacturer’s instructions, the purified DNA fragment was cloned into the vector pCR 2.1-TOPO^®^ (TOPO TA Cloning^®^ Kits, Invitrogen, Carlsbad, CA, USA). The new plasmid was used to transform electrocompetent *E. coli* XL1Blue cells (1.8 kV). The transformed cells were grown in Petri dishes on 2xYT agar supplemented with ampicillin (200 μg/mL) and incubated overnight at 37 °C.

Some colonies were selected and evaluated using colony PCR with the M13 forward and reverse oligonucleotides (Table 2). Plasmids from positive colonies with the expected DNA size were purified and sequenced at the Institute of Biotechnology, UNAM. The resulting sequences were searched for homology with the *P. depilata* transcriptome database.

### 5.5. In Vivo Biological Activities

#### 5.5.1. Acute Toxicity in Mice

The acute toxicity of the recombinant peptide purified by RP-HPLC was determined according to OECD Guideline 423 titled “Acute Oral Toxicity—Acute Toxic Class Method” [32], with some modifications. The test’s focus was to use the minimum number of animals per dosis (3 Swiss Webster, nulliparous mice, healthy female albino, not pregnant, 18–24 g, dosed by intracranial (ic) injection), but enough to obtain the test substance’s acute toxicity to determine its lethal dose 50 (LD_50_). The initial dose of the recombinant peptide was 0.9 µg/g, based on the LD_50_ of the *P. depilata* venom. Depending on the number of deaths, the corresponding dose was adjusted according to the following procedure: the dose was decreased if two to three deaths occurred, the dose was repeated to confirm the lethal dose, or the dose was increased if there were no deaths or only one death. The procedure was continued until the lethal dose was found.

After dosing, the mice were observed continuously for 30 min, then hourly for the first four hours and then every 24 h, until day 14, with deaths, signs of toxicity and body weights recorded daily. On day 14, the surviving animals were euthanized by CO_2_ over-exposure. Immediately after death, by poisoning or euthanasia, all mice underwent macroscopic examination of the major organs (heart, lungs, stomach, spleen, liver and kidneys) to assess their appearance and possible involvement. The result obtained was expressed in terms of the corresponding LD_50_. The study was conducted according to the tenets of the Declaration of Helsinki and approved by the Ethical Committee for Animal Experimentation of the Universidad de Antioquia in Act No. 104 of 15 June 2016.

#### 5.5.2. Toxicity and Mean Paralyzing Dose (PD_50_) of the Recombinant Peptide Ctx-4 in Crickets

The recombinant Ctx-4 purified from RP-HPLC was tested for toxicity in house crickets (*Acheta domesticus*, 0.1–0.2 g) using lateroventral intrathoracic injection (lit). The median paralyzing dose (PD_50_) in crickets was defined as the amount of peptide that caused paralysis in 50% of the crickets tested (20 crickets). The median lethal dose (LD_50_) was the amount of peptide that caused the death in 50% of the treated population. The PD_50_ and LD_50_ were determined according to the Dixon method [33]. Briefly, a single cricket was treated periodically with a fixed dose. If the first cricket was paralyzed for at least 1 min within the first 10 min after the injection, the next cricket was injected with a lower dose. Similarly, if the first cricket died 30 min after injection, the next was inoculated with a lower dose. This procedure continued until the required number of insects was dosed for calculating either the PD_50_ or the LD_50_. Mean values and confidence intervals were calculated [33]. The study was conducted according to the tenets of the Declaration of Helsinki and was approved by the Animal Experimentation Ethics Committee of the Universidad de Antioquia in Act No. 104 of June 15th, 2016.

## Figures and Tables

**Figure 1 toxins-15-00436-f001:**
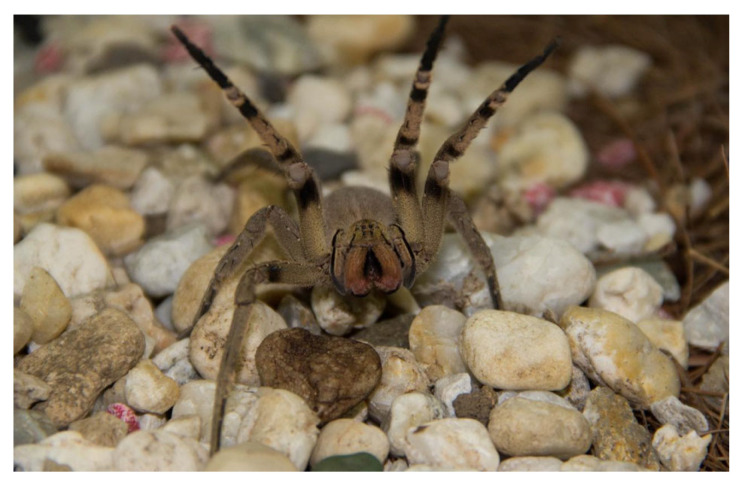
*Phoneutria depilata* (photo from the spider vivarium of the Serpentario at the Universidad de Antioquia, Medellín, Colombia).

**Figure 2 toxins-15-00436-f002:**
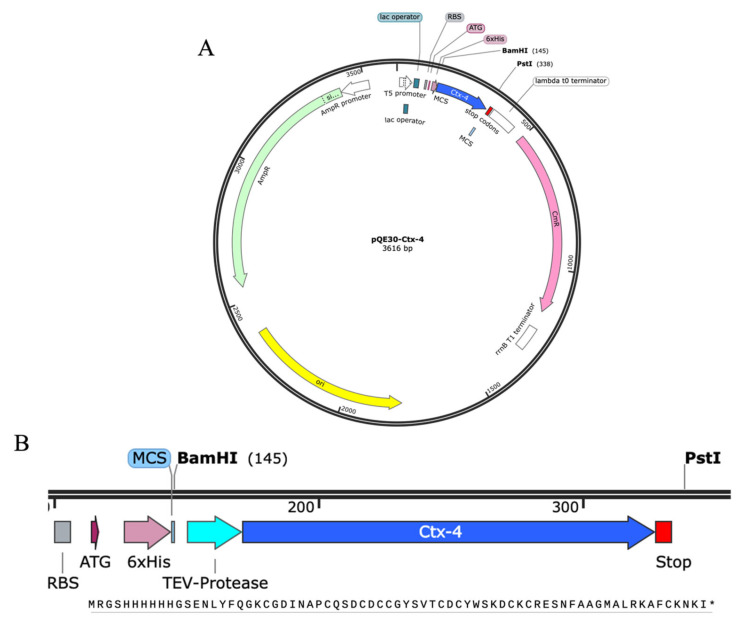
(**A**) Expression plasmid map pQE30::Ctx-4 and all its features. (**B**) The expression cassette: RBS, ATG, 6xHis, TEV protease site, Ctx-4 sequence, stop and restriction enzymes sites. Molecular size of expressed cassette 8006.03 Da (reduced protein).

**Figure 3 toxins-15-00436-f003:**
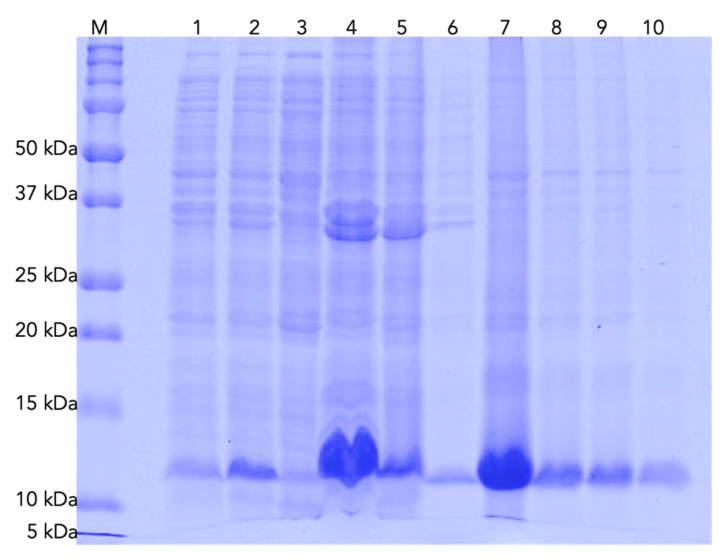
SDS-PAGE (4/15% Tris/Glycine). *E. coli* Origami cells harboring the plasmid pQE30::Ctx-4: expression and purification through affinity chromatography of rCtx-4. Wells: **M**. Molecular weight marker; **1**. Cells before induction; **2**. Cells after induction; **3**. Soluble fraction. **4**. Insoluble fraction. **5**. Eluate. **6**. Wash 15 mM Imidazole; **7**–**10**. Elution 400 mM Imidazole.

**Figure 4 toxins-15-00436-f004:**
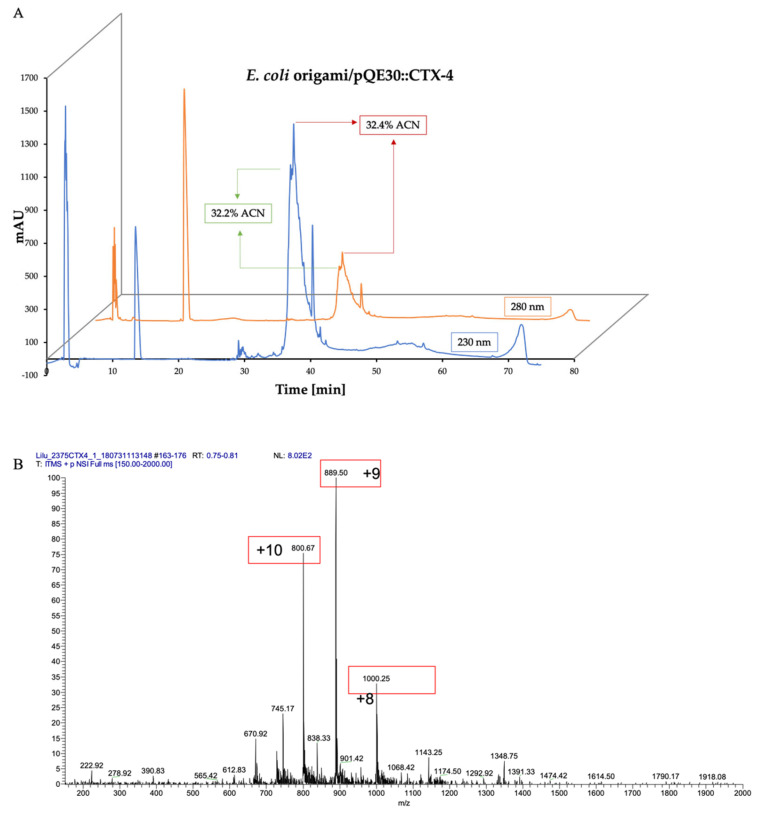
(**A**) RP-HPLC of the rCtx-4 peptide from nickel resin purification: 0–60% acetonitrile in 0.1% TFA in a linear gradient from minute 5 to minute 65 (1 mL/min). The run was monitored at 230 nm and 280 nm. (**B**) Mass spectrum of the rCtx-4 peptide. Experimental oxidized mass: 7995.73 Da. Theoretical oxidized mass 7995.7 Da. (Ion mass/charge: [+10] 7996.7, [+9] 7996.5, [+8] 7994. Average: 7995.7 Da).

**Figure 5 toxins-15-00436-f005:**
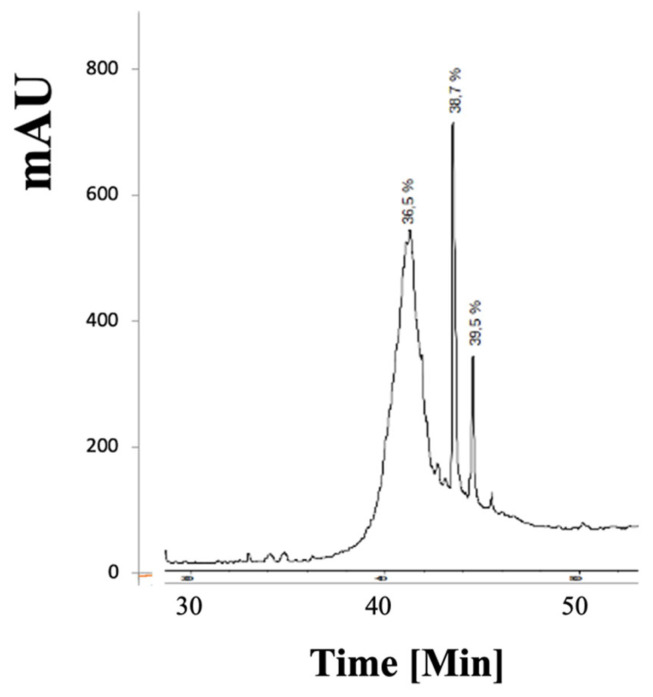
RP-HPLC purification of the folded rCtx-4 peptide. The run was monitored at 230 nm. The numbers at the top of each peak represent the ACN percentage for elution.

**Figure 6 toxins-15-00436-f006:**
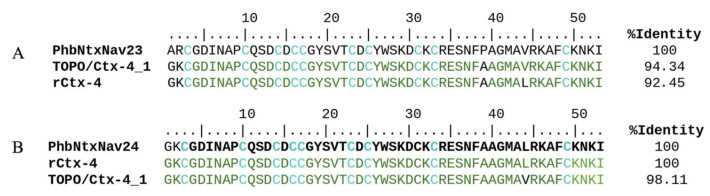
(**A**) Blastn alignment of the peptide sequences of the PhbNtxNav23 transcript, plasmid pCR 2.1-TOPO-Ctx and the recombinant Ctx-4. (**B**) Blastx alignment of the peptide sequences of the PhbNtxNav24 transcript, the rCtx-4 and plasmid pCR 2.1-TOPO-Ctx. Amino acid matches are shown in green.

**Figure 7 toxins-15-00436-f007:**
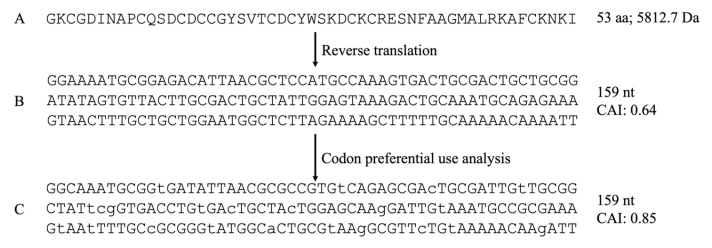
Gene design and construction of Ctx-4. (**A**) Amino acid sequence of transcript PhdNtxNav24. (**B**) Reverse-translation of Ctx-4. (**C**) Nucleotide sequence with preferential codon usage for expression of Ctx-4 in *E. coli* (lowercase letters denote changes upon preferential codon usage).

**Table 1 toxins-15-00436-t001:** Signal peptide, propeptide and mature peptide of the transcript PhdNtxNav24 [13].

Name	Amino Acid Sequence	Features
Signal peptide	MKVAIFFILSLFVLAVAS	18 amino acids
Propeptide	ESIEEKREEFPVEESAR	17 amino acids
Mature toxin	GKCGDINAPCQSDCDCCGYSVTCDCYWSKDCKCRESNFAAGMALRKAFCKNKI	53 amino acids

**Table 2 toxins-15-00436-t002:** Oligonucleotides used for construction of the gene coding for Ctx-4 and other primers.

Name	Sequence (5′ → 3′)	Features
Ctx_Up1	GAGA**GGATCC**GAGAACCTGTACTTTCAAGGCAAATGCGGtGATATTAACGCGCCGTGTCAGAG	63 nucleotides **GGATCC**: *Bam*HI site GAGAACCTGTACTTTCAA: TEV protease site
Ctx_Lw2	GCTCCAGTAGCAGTCACAGGTCACCGAATAGCCGCAACAATCGCAGTCGCTCTGACACGGCGCGTTA	67 nucleotides
Ctx_Up3	CCTGTGACTGCTACTGGAGCAAGGATTGTAAATGCCGCGAAAGTAATTTTGCCGCGGGTATGGCAC	66 nucleotides
Ctx_Lw4	TCTC**CTGCAG**CTATTAAATCTTGTTTTTACAGAACGCCTTACGCAGTGCCATACCCGCGGC	61 nucleotides **CTGCAG**: *Pst*I site CTATTA: stop codons
cDNA-Ctx-Fwr	AAATGCGGCGATATAAACG	19 nucleotides
cDNA-Ctx-Rev	TTATTATATTTTGTTTTTGCAGAAGGC	27 nucleotides
pQE-Fwd	GAGCGGATAACAATTATAA	19 nucleotides
pQE-Rev	GGTCATTACTGGATCTAT	18 nucleotides
M13-Forward	GTAAAACGACGGCCAGT	17 nucleotides
M13-Reverse	CAGGAAACAGCTATGAC	17 nucleotides

## Data Availability

Not applicable.

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
