# Peer review of "Heterologous Expression of an Insecticidal Peptide Obtained from the Transcriptome of the Colombian Spider Phoneutria depilate"

_toxins, 2023, doi:10.3390/toxins15070436_

Round 1

Reviewer 1 Report

While synthetic insecticides have been instrumental in controlling insect pests, they raise significant concerns not limited to environmental and human health concerns, non-target effects, and the development of resistance. The development of new bioinsecticides is driven by the need to address these concerns. By finding new bioinsecticides, we can reduce reliance on chemical insecticides and minimize the potential risks they pose to ecosystems, non-target organisms, and even humans. In their manuscript “Heterologous expression of an insecticidal peptide obtained from the transcriptome of the Colombian spider Phoneutria depilate” the authors present the primary structure of an insecticidal peptide from the venom gland transcriptome of the ctenid spider Phoneutria depilate. The recombinant peptide rCtx-4 exhibits significant toxicity in crickets but not in mice, suggesting the potential application of rCtx-4 as a bioinsecticide for controlling insect pests in agricultural and industrial settings.

Overall, the author’s preliminary findings address an important research gap—developing bioinsecticides to reduce reliance on chemical insecticides and contribute to integrated pest management strategies. Additionally, the authors have demonstrated that recombinant technology presents a significant opportunity for rapid and effective production of bioinsecticides in large quantities, vital for sustainable pest management strategies. Such technologies can contribute to the widespread adoption and availability of bioinsecticides which is vital in addressing the concerns posed by synthetic insecticides. 

While the work is satisfactory and highlights the promise of finding sustainable and effective solutions to the concerns posed by synthetic peptides, the authors need to address a few issues for the work to be suitable for publication.

1.      Method of application:  The authors have demonstrated that the peptide rCtx-4 exhibits significant toxicity in crickets following intrathoracic injection. However, it is important to note that most insecticides are applied in various formulations and delivery systems such as sprays, baits, slow-release diffusion, and others. Besides injection, the authors should evaluate whether traditional methods of insecticide application have any impact or significant impact on insect pests.

2.      Target specificity: While the findings that rCtx-4 peptide exhibits selective toxicity to crickets and not mice, the authors should evaluate the toxicity of the peptide on other equally important beneficial insects such as bees and butterflies

3.      In this article, the author applied homology search (NCBI-BLAST) to obtain a potential bioactive peptide sequence from the transcriptome data of spider P. depilate. I think the author should present the whole sequence (as supplementary data) of the deduced peptide including signal peptide, propeptide, and mature peptide in the article to make it more convincing.

4.      The connection modes of the disulfide bonds of the synthesized peptide need to be clarified.

5.      In the discussion part, the author hypothesized that rCtx-4 sequence may not affect sodium channels in mammals but in invertebrates. The authors should compare the differences between the sodium channels in mammals and invertebrates and explain their reasoning. I suggest the authors should also test and compare the bioactivities of rCtx-4 on different sodium channels in mammals or invertebrates to verify the hypothesis.

6.      The toxicity evaluation of rCtx-4 on mice needs to be reconsideration since body weight and organ appearance are rough indications. The author may consider adding tissue sectioning and inflammatory factors to make the results more consolidated.

Author Response

We would like to thank the reviewer for his/her time and comments on our paper. We will try to respond to all queries, and we hope our answers can fulfill the reviewer’s concerns.

  1. Method of application:  The authors have demonstrated that the peptide rCtx-4 exhibits significant toxicity in crickets following intrathoracic injection. However, it is important to note that most insecticides are applied in various formulations and delivery systems such as sprays, baits, slow-release diffusion, and others. Besides injection, the authors should evaluate whether traditional methods of insecticide application have any impact or significant impact on insect pests.

The reviewer is entirely correct, and we thank his/her appreciation. Although we would like to continue with the research, we would like to point out here that this is the first approach to the insecticidal activity of rCtx-4, and the administration route has not yet been tested. Due to its chemical characteristics and conditions, the evaluation of a delivery matrix for the peptide should be conducted first to ensure its stability, which will take further investigation out of the objectives of this study.

  1. Target specificity: While the findings that rCtx-4 peptide exhibits selective toxicity to crickets and not mice, the authors should evaluate the toxicity of the peptide on other equally important beneficial insects such as bees and butterflies

We understand the reviewer’s concern, and in future research with this bioinsecticide, we will consider the evaluation of beneficial insects, such as bees and butterflies. Meanwhile, in this research, we have conducted experiments following regulations for experimentation on animals provided by an ethics committee under the tenets of the Declaration of Helsinki. We performed toxicity assays according to the granted permits, which is why we did not use other insects.

  1. In this article, the author applied homology search (NCBI-BLAST) to obtain a potential bioactive peptide sequence from the transcriptome data of spider P. depilate. I think the author should present the whole sequence (as supplementary data) of the deduced peptide including signal peptide, propeptide, and mature peptide in the article to make it more convincing.

We thank the reviewer for his/her intentions to improve our manuscript. Indeed, we have added “Table 1” with the information on the signal peptide, pro-peptide, and mature peptide in the paper.

  1. The connection modes of the disulfide bonds of the synthesized peptide need to be clarified.

We thank the reviewer for his/her suggestions. As indicated in our paper, we obtained three fractions of rCtx-4 after folding (36.5, 38.7, and 39.5 % ACN). Fraction 36.5 % showed weak activity over crickets; meanwhile, 38.7 and 39.5 % ACN demonstrated an evident bioinsecticide activity. The results indicated that the obtained isoforms at 38.7 and 39.5 % ACN probably had a minimum energy structure like the native conformation, allowing us to consider an adequate disulfide pattern. In future research, it would be illustrative to elucidate how to join cysteines in rCtx-4.

  1. In the discussion part, the author hypothesized that rCtx-4 sequence may not affect sodium channels in mammals but in invertebrates. The authors should compare the differences between the sodium channels in mammals and invertebrates and explain their reasoning. I suggest the authors should also test and compare the bioactivities of rCtx-4 on different sodium channels in mammals or invertebrates to verify the hypothesis.

We thank the reviewer for the recommendation. As mentioned in our paper, we hypothesized that rCtx-4 might not affect mammalian sodium channels but invertebrates’ sodium channels. Our hypothesis was based on two antecedents. The first was the experimental procedures we conducted, where mice (mammals) were not affected by the recombinant toxin, but crickets (insects) were severely affected (dead) by the same recombinant toxin. The second antecedent was provided by other studies that have indicated that several arthropods’ toxins are selectively active on insect sodium channels but not on their mammalian counterparts (Bosmans et al. 2005) or that at a concentration of 1 µM of arthropod toxin, the insect-voltage-gated sodium channel, was profoundly modulated while its mammalian counterpart, the rat brain Na(v)1.2 channel, was not affected (Strugatsky, et al., 2005). The difference between the pharmacological characteristics of insect NaV channels and those of mammals allowed the construction of specific molecules that act on insect NaVs with virtually no effect on mammalian NaVs (Romanova, et al., 2022).

Bosmans F, Martin-Eauclaire MF, Tytgat J. The depressant scorpion neurotoxin LqqIT2 selectively modulates the insect voltage-gated sodium channel. Toxicon. 2005 Mar 15;45(4):501-7. doi: 10.1016/j.toxicon.2004.12.010. Epub 2005 Jan 20. PMID: 15733572.

Strugatsky D, Zilberberg N, Stankiewicz M, Ilan N, Turkov M, Cohen L, Pelhate M, Gilles N, Gordon D, Gurevitz M. Genetic polymorphism and expression of a highly potent scorpion depressant toxin enable refinement of the effects on insect Na channels and illuminate the key role of Asn-58. Biochemistry. 2005 Jun 28;44(25):9179-87. doi: 10.1021/bi050235t. PMID: 15966742.

Romanova, D.Y.; Balaban, P.M.; Nikitin, E.S. Sodium Channels Involved in the Initiation of Action Potentials in Invertebrate and Mammalian Neurons. Biophysica 2022, 2, 184-193. https://doi.org/10.3390/biophysica2030019

  1. The toxicity evaluation of rCtx-4 on mice needs to be reconsideration since body weight and organ appearance are rough indications. The author may consider adding tissue sectioning and inflammatory factors to make the results more consolidated.

We thank the reviewer for his/her comments. We would like to clarify that we assessed the toxicity according to the OCDE guideline for acute toxicity. This guideline does not require tissue sectioning, inflammatory factors, or a histopathological study; however, this could be considered when the experiment was performed. As this type of experiment requires the approval of an ethics committee, there are insufficient grounds to obtain permission to conduct a new toxicity test for this particular peptide.

Reviewer 2 Report

Key Contribution

Line 20 -> "...potential use industrial use in..." -> potential industrial use in -> review and correct;

Line 21 -> "...for it is nontoxic..." -> as it is nontoxic -> review and correct;

Introduction

Lines 24,29,54 -> no comma before and in these situations -> review and correct;

Line 55 -> "specie" -> species - "specie" does not exist in english -> correct;

Line 72 -> "...spider P. depilata. validating..." -> review punctuation mark;

Results

Line 101 -> "16 ºC" -> 16 °C -> correct;

Line 125 -> "rCtx-4. were" -> review punctuation mark;

Line 131 -> "...percentage ACN of its elution." -> ACN percentage for elution -> review;

Discussion

Line 207 -> "...with the same number of cysteines,..." -> What number? To which peptide, Ctx-4? -> It is not clear, please review;

Line 276 -> "...but the differences may be due to the binding pattern of disulfide bonds formed during folding." -> Is there any report in the literatura supporting this proposition? Please cite;

Line 277 -> Extra space in rCtx- 4 -> correct;

Line 296 -> "...similar in both invertebrates and in mammals." -> Why do you include mammals here, to explain/support your hypothesis? Was peptide δ-ctenitoxin-Pn1a tested in mammals? Please review;

Conclusion

You propose agro-industrial use of rCTx-4 and it should be interesting if you could add to discussion something about toxic effets of this type of venom to leafhopper or other type of agricultural pests.

Materials and Methods

Line 331 -> "gen" -> gene -> correct;

Line 337 -> "C-" -> C) -> standardize;

Line 362 -> 37ºC -> 37 °C -> standardize;

Line 373 -> 2xTY -> 2xYT -> correct;

Line 375 -> 2xTY -> 2xYT -> correct;

Line 391 -> GSSG; GSH -> introduce abbreviations for enzyme and glutatione; also cite manufacturer;

Line 399 -> "...were extracted, immersed, and homogenized in TRIzol..." -> review sentence;

Line 411 -> "...buffer, and 1 U..." -> no comma in this case;

Line 431 -> It should be important to state somewhere in text that no deaths were observed in rCtx-4 acute toxicity tests performed with mice!

Line 454 -> How many crickets were tested? Please mention;

Information about funding and acknowledgments are lacking!

English quality is fine.

Author Response

We would like to thank the reviewer for his/her time and comments on our paper. We will try to respond to all queries, and we hope our answers can fulfill the reviewer’s concerns.

Comments and Suggestions for Authors

Key Contribution

Line 20 -> "...potential use industrial use in..." -> potential industrial use in -> review and correct;

Corrected

Line 21 -> "...for it is nontoxic..." -> as it is nontoxic -> review and correct;

Corrected

Introduction

Lines 24,29,54 -> no comma before and in these situations -> review and correct;

Corrected

Line 55 -> "specie" -> species - "specie" does not exist in english -> correct;

Corrected

Line 72 -> "...spider P. depilata. validating..." -> review punctuation mark;

Corrected

Results

Line 101 -> "16 ºC" -> 16 °C -> correct;

Corrected

Line 125 -> "rCtx-4. were" -> review punctuation mark;

Corrected

Line 131 -> "...percentage ACN of its elution." -> ACN percentage for elution -> review;

Corrected

Discussion

Line 207 -> "...with the same number of cysteines,..." -> What number? To which peptide, Ctx-4? -> It is not clear, please review;

Corrected

Line 276 -> "...but the differences may be due to the binding pattern of disulfide bonds formed during folding." -> Is there any report in the literatura supporting this proposition? Please cite;

Corrected and cite added.

Line 277 -> Extra space in rCtx- 4 -> correct;

Corrected

Line 296 -> "...similar in both invertebrates and in mammals." -> Why do you include mammals here, to explain/support your hypothesis? Was peptide δ-ctenitoxin-Pn1a tested in mammals? Please review;

δ-ctenitoxin-Pn1a was also tested in mammals and there were no detected effects.

Conclusion

You propose agro-industrial use of rCTx-4 and it should be interesting if you could add to discussion something about toxic effets of this type of venom to leafhopper or other type of agricultural pests.

We understand the reviewer’s concern, and in future research with this bioinsecticide, we will consider the evaluation of other insects, such as bees, leafhoppers or butterflies. Meanwhile, in this research, we have conducted experiments following regulations for experimentation on animals provided by an ethics committee under the tenets of the Declaration of Helsinki. We performed toxicity assays according to the granted permits, which is why we did not use other insects.

Materials and Methods

Line 331 -> "gen" -> gene -> correct;

Corrected

Line 337 -> "C-" -> C) -> standardize;

Corrected

Line 362 -> 37ºC -> 37 °C -> standardize;

Corrected

Line 373 -> 2xTY -> 2xYT -> correct;

Corrected

Line 375 -> 2xTY -> 2xYT -> correct;

Corrected

Line 391 -> GSSG; GSH -> introduce abbreviations for enzyme and glutatione; also cite manufacturer;

Corrected

Line 399 -> "...were extracted, immersed, and homogenized in TRIzol..." -> review sentence;

Corrected

Line 411 -> "...buffer, and 1 U..." -> no comma in this case;

Corrected

Line 431 -> It should be important to state somewhere in text that no deaths were observed in rCtx-4 acute toxicity tests performed with mice!

Thank you for your suggestion. However, in the results section, numeral 2.5, we stated: “None of the mice treated (via ic) at doses of 0.9 μg/g of the folded rCtx-4 peptide showed signs of toxicity.”

Is it ok?

Line 454 -> How many crickets were tested? Please mention;

Corrected

Information about funding and acknowledgments are lacking!

Corrected

Comments on the Quality of English Language

English quality is fine.

Thank you, we appreciate your concept

Reviewer 3 Report

Animal venoms are rich sources of pharmacologically active substances. Among them, spider venoms are intensively studied for peptides with insecticidal activity, which represent an environmentally friendly alternative to conventional agrochemicals. The present manuscript describes a heterologous expression of an insecticidal peptide obtained from the transcriptome of the Colombian spider Phoneutria depilate. The peptide, which is 55 amino acids long and contains 10 Cys residues, was successfully folded, purified, and was active against crickets with an LD50 of 1.2 ug/g insect. However, at concentrations of 0.9-1.5 ug/g it was not toxic to mice and could therefore be used as a bioinsecticide.

The manuscript is well-written and the results are clearly presented and discussed. Before approving it for publication, I would like to ask the authors to correct some errors.

Line 20: A 'use' should be deleted in this sentence.

Line 55: Change 'specie' with 'species'.

Lines 71-73: This sentence needs to be corrected. You cannot find a gene in a cDNA library.

Lines 78-81: I suggest rewording this sentence as follows:

The transcript named Ctx-4 was also selected for recombinant expression because of its e-value (9e-21) and its content of cysteines (10), which form 5 disulfide bonds typical of insecticidal peptides from the venom of ctenid spiders [14].

Line 110: Does '4/15 % Glycine' mean '4-15% Tris/Glycine gel'? This should be corrected.

Lines 147-152: The term homology has been used, whereas the term identity should be used. Here you are comparing the identity and/or similarity of protein primary structures. Homology refers to common evolutionary origins.

Line 162: Change ‘has’ with ‘is’.

Lines 167, 176: Correct here and in the rest of the manuscript 'in mice' and 'in cricket' instead of 'on mice' and 'on cricket'.

Lines 171-174: I suggest changing the sentence as follows:

Crickets injected with the 36.5% ACN fraction (Figure 5) showed signs of paralysis one minute after administration, whereas injection of the 38.7 and 39.5% ACN fractions resulted in immediate paralysis of the crickets.

Line 179: Delete 'lit'.

Lines 186-187: This sentence would sound better if it were changed as follows:

'Crickets injected with the 38.7% and 39.5% ACN fractions died 29 and 32 hours after administration, respectively.'

Line 210: Change 'has' with 'was'.

Lines 253-254: Add the position numbers of the mutant amino acids leucine and valine.

Line 273: The abbreviation 'icv' is redundant and should be deleted. Change 'via intracerebroventricular' with 'after via intracerebroventricular administration'.

I have written many comments and suggestions in Comments to Authors.

Author Response

We would like to thank the reviewer for his/her time and comments on our paper. We will try to respond to all queries, and we hope our answers can fulfill the reviewer’s concerns.

Line 20: A 'use' should be deleted in this sentence.

Corrected

Line 55: Change 'specie' with 'species'.

Corrected

Lines 71-73: This sentence needs to be corrected. You cannot find a gene in a cDNA library.

Corrected

Lines 78-81: I suggest rewording this sentence as follows:

The transcript named Ctx-4 was also selected for recombinant expression because of its e-value (9e-21) and its content of cysteines (10), which form 5 disulfide bonds typical of insecticidal peptides from the venom of ctenid spiders [14].

Corrected

Line 110: Does '4/15 % Glycine' mean '4-15% Tris/Glycine gel'? This should be corrected.

Corrected

Lines 147-152: The term homology has been used, whereas the term identity should be used. Here you are comparing the identity and/or similarity of protein primary structures. Homology refers to common evolutionary origins.

Corrected

Line 162: Change ‘has’ with ‘is’.

Corrected

Lines 167, 176: Correct here and in the rest of the manuscript 'in mice' and 'in cricket' instead of 'on mice' and 'on cricket'.

Corrected

Lines 171-174: I suggest changing the sentence as follows:

Crickets injected with the 36.5% ACN fraction (Figure 5) showed signs of paralysis one minute after administration, whereas injection of the 38.7 and 39.5% ACN fractions resulted in immediate paralysis of the crickets.

Corrected

Line 179: Delete 'lit'.

Corrected

Lines 186-187: This sentence would sound better if it were changed as follows:

'Crickets injected with the 38.7% and 39.5% ACN fractions died 29 and 32 hours after administration, respectively.'

Corrected

Line 210: Change 'has' with 'was'.

Corrected

Lines 253-254: Add the position numbers of the mutant amino acids leucine and valine.

Corrected

Line 273: The abbreviation 'icv' is redundant and should be deleted. Change 'via intracerebroventricular' with 'after via intracerebroventricular administration'.

Corrected